

# Comparison of conditioning factors classification criteria in large scale statistically based landslide susceptibility models

Marko Sinčić[1], Sanja Bernat Gazibara[1], Mauro Rossi[2], Snježana Mihalić Arbanas[1]

[1]Faculty of Mining, Geology and Petroleum Engineering, University of Zagreb, Pierottijeva 6, 10000 Zagreb, Croatia
[2]Consiglio Nazionale delle Ricerche, Instituto di Ricerca per la Protezione Idrogeologica, via Madonna Alta 126, 06128 Perugia, Italy

*Correspondence to*: Marko Sinčić (msincic@rgn.hr)

**Abstract.** The large scale landslide susceptibility assessment (LSA) is an important tool for reducing landslide risk through the application of resulting maps in spatial and urban planning. The existing literature more often deals with LSA modelling
techniques and the scientific research very rarely focuses on acquiring relevant thematic and landslide data, necessary to achieve reliable results. Therefore, the paper focuses on the crucial step of classifying continuous landslide conditioning factors for susceptibility modelling by presenting an innovative comprehensive analysis that resulted in 54 landslide susceptibility models to test 11 classification criteria (scenarios which vary from stretched values, partially stretched classes, heuristic approach, classification based on studentized contrast and landslide presence, and commonly used classification criteria, such
as Natural Neighbor, Quantiles and Geometrical intervals) in combination with five statistical methods. The large scale landslide susceptibility models were derived for small and shallow landslides in the pilot area (21 km$^2$) located in the City of Zagreb (Croatia), which occur mainly in soils and soft rocks. Some of the novelties in LSA are the following: scenarios using stretched landslide conditioning factor values or classification with more than 10 classes prove more reliable; certain statistical methods are more sensitive to the landslide conditioning factor classification criteria than others; all the tested machine learning
methods give the best landslide susceptibility model performance using continuously stretched landslide conditioning factors derived from high-resolution input data. The research highlights the importance of qualitative assessments, alongside commonly used quantitative metrics, to verify spatial accuracy and to test the applicability of derived landslide susceptibility maps for spatial planning purposes.

## 1 Introduction

Addressing landslide hazard is commonly done by zoning, i.e. deriving landslide susceptibility, hazard, and risk zoning maps (Corominas et al., 2013). Brabb, 1984 defined landslide susceptibility as a 'likelihood of a landslide occurring in a given area', indicating the spatial component exclusively. In the years to come, Soeters and van Westen, 1996, Guzzeti et al., 1999, van Westen et al., 2008, Fell et al., 2008a,b, Corominas et al., 2013, and Reichenbach et al., 2018 represent some of the most significant progress done in the field of research considering landslide susceptibility. One of the most popular approaches to
derive landslide susceptibility models (LSM) is using statistical methods, where the most recent and detailed review of





statistically based landslide susceptibility models is given in Reichenbach et al., 2018, whereas Merghadi et al., 2020 emphasize only machine learning methods by reviewing algorithm performance. Literature (e.g. van Westen et al., 2008 and Corominas et al., 2013) has shown two main groups of data needed for applying statistical methods in landslide susceptibility assessments (LSA), i.e. landslide inventory maps and landslide conditioning factors (LCFs). Moreover, Reichenbach et al.,

2018 identified acquiring relevant landslide and thematic information (i.e. LCF) as the first two steps for preparing a LSA. Many papers discuss different methods (e.g. Wang et al., 2016a, Chen et al., 2017, Merghadi et al., 2020), mapping units (e.g. Bornaetxea et al., 2018, Jacobs et al., 2020), inventory types (e.g. Guzzetti et al., 2012, Petschko et al., 2015) and the importance and/or selection of LCFs (Donati and Turrini, 2002, Jebur et al., 2014, Gaidzik and Ramírez, 2021), which are all necessary steps in LSA according to Reichenbach et al., 2018.

This research is situated in the European Panonnian Basin, i.e. in the City of Zagreb in hilly regions of the southern foothills of Medvendica Mt., which are susceptible to sliding. Despite long-term investigation on landslide phenomena, Jurak et al., 1996 and Mihalić, 1998 identified the lack of sustainable landslide inventories and landslide hazard maps as the main issue in the landside risk management in the Republic of Croatia. As a result, a landslide inventory map (Bernat Gazibara et al., 2019a) and a landslide susceptibility map (Bernat Gazibara et al., 2023) were derived, followed by emphasizing the importance and

necessity of large scale landslide susceptibility maps in the system of spatial and urban planning (Mihalić Arbanas et al., 2023). Regardless of the degree of urbanization, landslide occurrence is commonly related to geomorphological, geological and climate settings, as well as anthropogenic factors. Furthermore, Bernat Gazibara et al., 2017 indicate that the main landslide triggers in the area are precipitation and snow melting, i.e. long continuous precipitation periods or short precipitation periods of high intensity. The input data for LSA in this study were successfully acquired during previous research investigations in

the study area (e.g. Bernat Gazibara et al., 2017, Bernat Gazibara et al., 2019a, Bernat Gazibara et al., 2023), and past large scale assessment in Croatia (Sinčić et al., 2022a), but their optimal application in landslide susceptibility modelling remained an open question. The relevance and new insight on acquiring input data for preparing LCFs on a large scale (i.e. 1:5 000) were provided by Sinčić et al., 2022a and proved by analyzing the predictive performances of large scale LSA by Krkač et al., 2023.

This relevant aspect of LSA was addressed by Jebur et al., 2014, which used Weight of Evidence (WoE), Logistic Regression (LR) and Support Vector Machine (SVM) methods in a large scale case study to compare a LiDAR derived LCF set with a set containing additional LCFs such as land use and geology. AUC comparison between the two scenarios favored LiDAR only derived LCFs, addressing the amount and type of LCFs used in the research, but not their classification criteria. Similarly, Dou et al., 2015 applied LR and Statistical Index methods to demonstrate that six LCFs with high correlation to landslide occurrence

result in better success and prediction rate than a complete set of 15 LCFs. Donati and Turrini, 2002 show that when using categorical LCFs, only a few classes can significantly influence the LSA, confirming the importance of how LCFs are classified. These studies showed the need for relevant LCF acquisition and selection criteria, whereas the Leave One Out test available in LAND-SUITE software (Rossi et al., 2022), certainty factor models (Dou et al., 2015), and variable importance (Shirvani, 2020) are some methods used for LCF selection.





LCF classification issues were discussed by Yan et al., 2019, who defined two classification criteria for LCFs selected by reviewing classification criteria in different studies available in the literature. Specifically, the two classification criteria for each of the five LCF results with fewer or more classes used to develop 32 LSM scenarios to test all the combinations. From the 32 derived maps, a low difference of 0.03 AUC between minimum and maximum AUC was identified. Namely, the best result was obtained when all LCFs were used in a scenario that included more classes in most LCFs. Xiao et al., 2021

implemented a different strategy, aimed to reduce local correlations among LCFs by reclassifying them to increase LSA accuracy. As stated by Bonham-Carter et al., 1990 and Neuhäuser and Terhorst, 2007, contrast or studentised contrast ($C_{st}$) metrics can be used to determine classification cut-off values in continuous LCFs when using a bivariate approach such as WoE or Information Value (IV). Moreover, generalizing continuous LCFs enables maximized spatial relations and statistical robustness (Neuhäuser et al., 2012). Concretely, Mathew et al., 2007 used the $C_{st}$ curve maximum to split the distance to roads,

drainage, and lineament LCFs into two classes. Furthermore, Neuhäuser et al., 2012 applied the $C_{st}$ curve in the WoE method to convert continuous LCFs to categorical by observing maximum and local maximum $C_{st}$ values. Jebur et al., 2014 used mainly the quantile criteria to derive ten classes for stretched rasters and a heuristic approach for buffer zones from vector lines (i.e. proximities or distances). On the other hand, Yusof et al., 2015 applied only the quantile criteria to classify LCFs into ten classes, based on the approach by Tehrany et al., 2013 and Tehrany et al., 2014 in flood susceptibility mapping. Huang

et al., 2020a,b and Huang et al., 2022 applied the natural breaks classification criteria and defined eight classes in each LCF. Wang et al., 2016b, Cui et al., 2017, Zhao and Chen, 2020, and Wang et al., 2020 used what seems to be equal intervals defined heuristically without specifications, whereas Wang et al., 2021 did not classify continuous LCFs and applied them as stretched rasters in the analysis. Several papers dealing with large scale LSA (e.g. Vojteková and Vojtek, 2020 and Xing et al., 2021) also do not specify the classification criteria but present the LCFs with popular equal interval classes likely defined

heuristically. Similarly, Wang et al., 2016b used equal intervals to classify slope and buffer zones for LCFs, whereas a heuristic approach was selected for stretched rasters. In most of the mentioned studies, which applied a heuristic approach, a relatively low number of classes in LCFs are often defined, i.e. from 4 to 6 classes. A detailed comparison of all these criteria and techniques is still missing, especially on large scale assessments and with adequate metrics. Based on the latter, and as confirmed by Huang et al., 2020b and Xing et al., 2021, there is no uniform approach to classifying continuous LCFs. It can

be concluded that no systematic analysis to compare the criteria was done, as the researchers are instead applying what was already confirmed successful in different individual case studies.

Few papers deal with LCF processing once they are selected, i.e. discussing the process of classifying continuous LCFs (e.g. Yan et al., 2019, Xiao et al., 2021), which is the scope of this study. Therefore, this paper compares 11 criteria used to define the processing of continuous LCFs, i.e. transform relevant input data layers into LCFs suitable for application in landslide

susceptibility modelling. Unlike Yan et al., 2019, which are more oriented into case study combinations of LCFs with different derived LCF classes, we aim to develop a uniform approach for classifying continuous LCFs, leading to defining a first step for large scale LSA methodology. Besides testing the 11 scenarios where continuous LCFs are categorized, each scenario is applied in five statistical landslide susceptibility methods, including IV, Logistic Regression (LR), Neural Network (NN),



Random Forests (RF), and SVM. Such methods were considered to analyze the influence of LCF classification criteria on LSA
accuracy using different statistical modelling approaches, and they were selected because they are commonly used in LSA
(Reichenbach et al., 2018).

Remote sensing has proven helpful for LSA, leading to better landslide hazard mitigation strategies. Concretely, different
remote sensing techniques are widely applied for data acquisition when studying landslide hazard, as presented by Scaioni et
al., 2014 and Ray et al., 2020, whereas in this paper, geomorphological and hydrological LCFs are based on LiDAR (Light
Detection and Ranging) point cloud data acquired with Airborne Laser Scanning (ALS). Furthermore, the landslide inventory
map was derived by mapping on morphometric maps derived from a high-resolution DTM (Digital Terrain Model), which
shows a significant advantage compared to other methods for mapping small landslides under vegetation (Razak et al., 2013,
Bernat Gazibara et al., 2019a). Jebur et al., 2014 argue that LiDAR-derived LCFs could be sufficient for LSA where geological
LCFs (e.g. soil and geology) are not available, whereas Sinčić et al., 2022a were able to derive a complete set of LCFs together
with a set of elements at risk for large scale LSA by using exclusively high-resolution LiDAR and orthophoto data in
combination with publicly available small scale geological data.

Besides a study case presented by Yusof et al., 2015 using LR and Evidential Belief Function, comparing LSA methods on a
large scale is not common in the literature. Hence, one of the novelties of this paper is presenting a detailed comparison,
including the quantitative and qualitative perspective of five popular LSA methods applied on a large scale with relevant input
data of high spatial accuracy. Zêzere et al., 2017 point out that the best AUC metric values do not necessarily define the best
LSM. Similarly, Vakhshoori and Zare, 2018 emphasize that AUC values might be deceiving or ineffective in detecting
uncertainties in spatial prediction and can only indicate general reliability. To address the issue, Vakhshoori and Zare, 2018
suggest studying additional metrics, such as Cohen's kappa, to acquire additional information about the LSM. Considering the
susceptibility quality level (SQL) introduced by Guzzetti et al., 2006a,b, Reichenbach et al., 2018 identified a low amount of
papers published with a high SQL rank. Addressing model fitting and predictive performance and measuring uncertainties in
the derived models, LSMs derived in this paper are considered high SQL, ensuring systematical and unbiased comparison.
Moreover, researching the application of LSMs in spatial planning (Mihalić Arbanas et al., 2023), we qualitatively present
derived LSMs in close up views and high resolution to measure the applicability of the used zonation method, i.e. determine
visually susceptibility class area distribution. However, choosing the optimal zonation method is out of the scope of this study,
and the one presented in this paper serves exclusively for uniform comparison of derived LSMs. Moreover, it should be stated
that optimization of used statistical methods is a topic beyond the objective of this research. We argue that large scale LSA
using high quality input data should also be measured qualitatively as expert judgement by observing real environmental
conditions on high resolution DTM derivatives. It provides necessary insight into map quality and applicability, which is not
detected by commonly used quantitative approaches (e.g., AUC).





## 2 Materials

### 2.1 Study area

The study area is located in the City of Zagreb, the capital of Croatia, in the northwest part of the country. It encompasses 21 km$^2$ of the southern slopes of the Medvednica mountain, i.e. the western part of the Podsljeme area. The City of Zagreb belongs to the European Pannonian Basin, whereas the Podsljeme area encompassing the study area is depicted with hilly relief, with 90% of the study area steeper than 5° (Bernat Gazibara et al., 2019b). Basic geological settings can be described as upper Miocene and Quaternary sediments making up 92% of the study area, described in detail in the geological map 1:100 000 and supplementary geological notes developed by Šikić et al., 1972 and Šikić et al., 1979, respectively. With a dense population and a high degree of urbanization, which is increasing, the Podsljeme area has been under research regarding landslide phenomena in the last 50 years, starting with the first landslide inventory map by Šikić, 1967, and following geomorphological landslide maps from 1979 (Polak et al., 1979) and a geomorphological inventory from 2006 (Miklin et al., 2006). More recently, Bernat Gazibara et al., 2014a,b described landslide events triggered by intensive rainfall. Moreover, Mihalić Arbanas et al., 2016 identified the lack of a suitable landslide inventory as one of the critical issues in landslide risk management in the City of Zagreb. As a result, a LiDAR based landslide inventory map for the 21 km$^2$ in this study area was derived (Bernat Gazibara et al., 2019a,b), followed by a susceptibility assessment (Bernat Gazibara et al., 2023). The LSA included a WoE method, which was initially used in the geologically and geomorphologically representative 21 km$^2$ pilot area to obtain weight values, which were further applied to a more extensive territory of the Podsljeme area. Satisfactory results were obtained, confirming the proposed methodology for large scale LSA. Considering the highly urbanised environment, population density and a continuous increase of human induced landslides in the Podsljeme area (Jurak et al., 2008 and Mihalić Arbanas et al., 2014), a large scale landslide susceptibility assessment is necessary for adequate landslide management in the study area.

### 2.2 Input data

The first landslide inventory map developed from LiDAR data for the study area was completed by Bernat Gazibara et al., 2019a,b based on ALS, which was performed during the leaf-off period in Croatia in 2013. Furthermore, another LiDAR ALS was performed in 2020 (Bernat Gazibara et al., 2022, Bernat Gazibara et al., 2023) to verify the existing landslide inventory map, providing a multi-temporal insight into landslide occurrence. Namely, the landslide inventory map consists of 702 mapped polygons, with the most frequent landslide area being 400 m$^2$ and a density of 33 landslides per square kilometre. The input data layers needed to derive LCFs are prepared from source data, as depicted in Table 1.



**Table 1 Overview of source data used for derivation of input data layers**

| Source Data | Scale | Input data layer | Obtained by |
|---|---|---|---|
| LiDAR point cloud | 5 m resolution | Elevation | Interpolation |
| | | Slope | ArcGIS 10.8 Slope tool |
| | | Aspect | ArcGIS 10.8 Aspect tool |
| | | Terrain wetness | Evans et al., 2014 |
| | | Drainage network | ArcGIS 10.8 Spatial Analyst Toolbox |
| Croatian Basic Geological Map | 1:100 000 | Lithology (rock type) | Digitization |
| | | Geological contact | Digitization |
| | | Faults | Digitization |
| Topographic Map of Croatia | 1:25 000 | Streams | Digitization |
| Land use planning maps | Large scale | Land use | City of Zagreb (2011) |


Besides the LiDAR point cloud, the Croatian Basic Geological Map (Šikić et al., 1972), the Topographic Map of Croatia (State Geodetic Administration) and the City of Zagreb land use planning maps (City of Zagreb, 2011) were used to complete the set of source data. We note that the used geological input data is small scale, but considering preliminary analysis in the LCF selection process, it resulted in being more relevant to landslide occurrences in terms of the leave one out test in comparison

to the alternative geological map in scale 1:5 000. Furthermore, the mentioned small scale geological input data was also applied in Bernat Gazibara et al., 2023 where it yielded excellent results. Upon deriving elevation (Fig. 1a) from the LiDAR point cloud using interpolation, ArcGIS 10.8 and Geomorphometry and Gradient Metrics (Evans et al., 2014) toolboxes were used on the elevation input data layer to derive slope (Fig. 1b), aspect (Fig. 1c), terrain wetness (Fig. 1e) and drainage network (Fig. 1f). Digitization enabled deriving geological input data layers (Fig. 1d) from the Croatian Basic Geological Map (Šikić

et al., 1972), as well as obtaining streams from a Topographic Map of Croatia (State Geodetic Administration). Lastly, land use maps from the City of Zagreb were reclassified accordingly and used as land use input data layer illustrated in Fig. 1g.

Input data layers can be classified according to their origin in continuous and categorical, as presented in Table 2. Furthermore, continuous input data layers can be subdivided into vector lines (e.g. geological contacts, faults, drainage network, all streams) and stretched rasters (e.g. elevation, slope, terrain wetness). On the other hand, lithology (rock type) and land use made up the

polygon group of categorical input data layers, unlike aspect being the only categorical raster type. Continuous stretched rasters are presented with edge values (Fig. 1a, b, e), whereas categorical input data layers are depicted with their classes (Fig. 1c, d, g). Furthermore, the continuous line input data layers' spatial presence is illustrated in Fig. 1d, e, f.



**Table 2 Categorization and terminology definition of input data layers**

| Input data layers | | | |
|---|---|---|---|
| **Continuous** | | **Categorical** | |
| **Line (vector)** | **Stretched raster** | **Polygon (vector)** | **Classified raster** |
| Geological contact<br>Faults<br>Drainage network<br>All streams | Elevation<br>Slope<br>Terrain wetness | Lithology (rock type)<br>Land use | Aspect |




**Figure 1 Input data layers for landslide susceptibility analyses: (a) elevation; (b) slope; (c) aspect; (d) lithology (soil/rock type), geological contact and faults; (e) terrain wetness and streams; (f) drainage network; (g) land use; (h) landslide dataset**





## 3 Methodology

### 3.1 Study workflow

Figure 2 synthesizes the methodology applied in this study that can be split into 'preparing input data', 'susceptibility analyses', and 'quantitative and qualitative analyses' steps. Namely, relevant landslide and thematic (input data layers) information were obtained as described in the previous Sect. 2. Preparing input data involves deriving continuous LCF sets applicable in 11 classification scenarios and defining a fixed landslide dataset. Susceptibility analyses includes deriving 54 LSMs using the selected mapping unit with five statistical methods applied with the 11 classified factors scenarios. The quantitative and qualitative analyses considered three directions: (i) model evaluation parameters, (ii) LSM validation parameters, and (iii) LSM classification parameters. Commonly used quantitative parameters are examined in all three directions, whereas the qualitative approach is made only for the LSM classification, focusing on the LSM applicability, i.e. observing spatial distribution of susceptibility classes and variability of susceptibility values. The described workflow ensures the development of landslide susceptibility maps of high quality rank according to the SQL (Guzzetti et al., 2006a,b). Lastly, the quantitative and qualitative analysis results were summarized, enabling drawing the conclusions.






**Figure 2 Landslide susceptibility modelling workflow**






## 3.2 Preparing input data

The polygon-based landslide inventory map was randomly split into two sets, each containing the same amount of polygons. Unstable training polygons (first set) were transformed into pixels, i.e. 5 m raster chosen as appropriate for LSA on a large scale done in this study. Unstable training pixels were subtracted from the study area extent, and then an equal amount of

stable pixels were randomly selected in the remaining territory. This ensured an unbiased landslide training set with an equal amount of stable and unstable pixels for deriving LSMs, which also defines the model training dataset for model evaluation parameters. An unstable set of polygons used for validation (second set) and all unstable polygons used for classification were transformed into 5 m rasters and used for determining LSM validation and classification parameters, respectively.

Categorical LCFs (aspect, lithology, land use) are equivalent to input data layers presented in Table 2 and were used equally

in all 11 scenarios, an exception being aspect in scenario S11, as explained further in the section. Classes in categorical LCFs were ordered according to frequency ratio values defined by the presence of an unstable training landslide set. Continuous LCFs were derived differently from input data layers (Table 2, Fig. 1), as follows for 11 scenarios.

Continuous stretched rasters (elevation, slope, terrain wetness) were reclassified into 100 equal classes for scenario S1. Regarding continuous line vector LCFs (geological contact, faults, drainage network, all streams), 100 equal multiple buffer

ring zones were derived to finalize scenario S1 LCFs (Fig. 3a). Similarly to scenario S1, scenario S2 is defined by 50 equal classes, as illustrated in Fig. 3b. The aim of defining scenarios S1 and S2 is to simulate the continuous input data layer to a high and low detailed extent, respectively. In Scenario S3, LCFs were defined by heuristically classifying and defining buffer zones for continuous rasters and line input data layers, respectively. The subjective approach was led by researchers' experience (e.g. Sinčić et al., 2022b, Krkač et al., 2023) and previous work in the study area (e.g. Bernat Gazibara et al., 2023).

For each of the 100 scenario S1 LCF classes, $C_{st}$ values were calculated based on the bivariate approach, i.e. observing class pixel size and landslide pixels presence in the class. Namely, $C_{st}$ is defined as a ratio of contrast (C) to its standard deviation s(C) (Bonham-Carter, 1994), i.e.:

$$C_{st} = \frac{c}{s(C)} \tag{1}$$

where,

$$C = W^+ - W^- \tag{2}$$

where $W^+$ and $W^-$ indicate positive and negative weight factors in the WoE method, respectively. Standard deviation s(C) is defined as:

$$s(C) = \sqrt{S(W^+)^2 + S(W^-)^2} \tag{3}$$

where $S(W^+)^2$ and $S(W^-)^2$ are variances of weights defined by Bishop et al., 1975. The $C_{st}$ curve was defined for each LCF,

followed by observing positive and negative $C_{st}$ trends throughout the 100 classes. Cut-off lines for scenario S4 were




determined by dividing the $C_{st}$ curve into positive and negative segments, as depicted in Fig. 3c. To define scenario S5, each scenario S4 positive segment was split into two by defining an additional cut-off line at the highest peak point (Fig. 3d).

**Figure. 3 Theoretical example of the studentized contrast and landslide density curves for the methodology applied to define cut-off values in scenarios S1 (a), S2 (b), S4 (c), S5 (d), S6 (e) and S7 (f), respectively**





Similarly, scenario S6 was defined with an additional cut-off line from scenario S5 at the lowest peak point of negative segments (Fig. 3e). For the theoretical example in Fig. 3c, d , e, the LCF for scenarios S4, S5 and S6 would result in 3, 4, and 6 classes, respectively. For scenario S7, the amount of expected and mapped landslide pixels was determined for each of the 100 classes previously defined in scenario S1. For stretched rasters (e.g. elevation and slope), expected landslides are defined

by the hypothesis stating that in each of 100 LCF classes the landslide density should be equal to the total study area landslide density, whereas the mapped landslide area was acquired by simple observation. After calculating the mapped and expected landslide area for each class, landslide density curves considering each class are constructed (Fig. 3f). Furthermore, the cut-off lines were determined by trend changes of landslide area presence, i.e. at the points where expected or mapped landslides change to being higher or lower than the other.

For an example given in Fig. 3f, the trends change three times resulting in 3 cut-off lines, i.e. four classes. On the other hand, the constant expected amount of landslides $A_{exp}$ for line vector LCFs used in scenario S7 was defined by:

$$A_{exp} = \frac{i}{i_{max}} \times A_{tot} \tag{4}$$

where i is the buffer interval size, $i_{max}$ the maximum reached buffer distance, and $A_{tot}$ total landslide area. The equation aims to define an equal expected landslide area in each buffer ring. Then, the area of mapped landslides in each buffer ring is

compared to the constant $A_{exp}$. Similarly, as with stretched rasters, trends in relations between mapped landslides being lower or higher than the constant $A_{exp}$ define the cut-off lines. It should be noted that a training landslide dataset was used for calculations needed to define scenarios S4-S7. Continuous stretched rasters and scenario S1 vector lines were reclassified using Natural Breaks (NB), Quantile (Q), and Geometrical Interval (GI) classification criteria. As a result, scenarios S8, S9 and S10 are developed, with each having ten classes, defined by NB, Q and GI reclassification criteria, respectively. Lastly, scenario

S11 was defined by using stretched continuous rasters without classifying them, i.e. as input data layers containing edge values rather than classes. Moreover, the aspect input data layer was applied with its original stretched values (i.e. 0-360°) representing a continuous LCF, unlike being categorical in the remaining ten scenarios. Lastly, scenario S1 line continuous LCFs with 100 equal classes were used in scenario S11 to simulate an input data layer as closely as possible. The presented classification criteria for 11 scenarios was applied uniformly, meaning each stretched raster and/or line vector was processed equally inside

each scenario, yet methodologically different from other scenarios. Scenarios for proposed LCF classification criteria varies from stretched (i.e. S11), partially stretched (i.e. S1 and S2), heuristic (i.e. S3), classified based on studentized contrast curve (i.e. S4, S5 and S6), classified based on expected and mapped landslide presence (i.e. S7), and lastly the commonly used classification criteria such as NB, Q and GI (i.e. S8, S9 and S10). The main aim is to test the stated classification criteria' relevance and determine its influence and necessity while using different statistical methods in a large scale case study.






### 3.3 Susceptibility analyses

As a first step in susceptibility analyses, LCF collinearity testing was performed individually for 11 scenarios in LAND-SUITE (Rossi et al., 2022). Correlations were examined by detecting LCF collinearity regarding Pearsons' R absolute value of 0.5 as
the cut-off threshold. Namely, values higher than 0.5 indicate collinearity between two examined LCFs, suggesting excluding one from the further susceptibility analyses. Selected LCFs showed no collinearity in all 11 scenarios. Furthermore, 54 LSMs were derived using the prepared landslide dataset and 11 LCF sets in the five selected methods, i.e. Information Value (IV), Logistic Regression (LR), Neural Network (NN), Random Forests (RF) and Support Vector Machine (SVM). It should be noted that the IV method was not applied in scenario S11, as described further below. The methods' optimization is out of the
scope of this study, as they are already widely known, and their usage is discussed by researchers in LSA studies. For a detailed theoretical background about the methods applied in this paper, the readers should read van Westen ,1993 and Merghadi et al., 2020.

IV is a simple bivariate statistical method developed by Yin and Yan, 1988. Based on landslide density, using the equation:

$$I_i = log \frac{S_i N}{N_i S} \tag{5}$$

where $S_i$ is the amount of landslide pixels in the observed class; $N_i$ the number of pixels in the observed class; $S$ the number of landslide pixels used for model training; $N$ amount of pixels in the study area and $I_i$ the information value of the observed variable. Positive and negative $I_i$ values indicate instability and stability, respectively, whereas higher values indicate a stronger relationship. Detailed methodology for using the bivariate statistical approach in landslide susceptibility analyses is given in van Westen, 2002, whereas IV is applied by Sarkar et al., 2013, Farooq and Akram, 2021 and Krkač et al., 2023 in
different LSA.

To achieve posterior probability, i.e. probabilistic [0,1] susceptibility values, the numerical $I_i$ is converted using the equation:

$$y = \frac{e^{f(x)}}{1+e^{f(x)}} \tag{6}$$

where: $f(x)$ is the numerical susceptibility value (input) and $y$ the probabilistic susceptibility value (output) as defined by Bonham-Carter, 1994. The described equation has a 0.5 cut-off value, defining <0.5 and >0.5 values as stable and unstable,
respectively.

IV is the only method in this study using only unstable modelling pixels for training, whereas other methods also require the randomly generated stable pixels. Consequently, IV LSM is not derived for scenario S11 as IV only applies to LCFs with classes, i.e., not compatible with continuous stretched input data layers that define scenario S11. On the other hand, scenario S1 with 100 equal classes approximates scenario S11 and can be considered as the closest alternative.
Introduced with the early work of Cox, 1958, LR today corresponds to the most common statistical classification method used in LSA (Reichenbach et al., 2018), as seen in the study cases from Rossi et al., 2010, Hemasinghe et al., 2018, and Bornaetxea et al., 2018, with practical code in a software application is available in LAND-SUITE (Rossi et al., 2022). A linear fitting function ($Z$) between landslides for $n$ amount of conditioning variables is defined with the equation:





$$Z = b_0 + b_1 X_1 + b_2 X_2 + \cdots + b_n X_n \tag{7}$$

where: $b_0$ is the intercept of the model, $b_n$ the partial regression coefficients, and $X_n$ is the conditioning variable. Lastly, a common application of the LR method includes converting resulting $Z$ values to probabilistic output by applying equation [6] as mentioned in Bornaetxea et al., 2018 and Merghadi et al., 2020.

NN is a two-stage regression or classification model that can handle multiple quantitative responses (Hastie et al., 2009). Among artificial NN methods, in this study feedforwarding was applied, indicating the flow of information exclusively in one

direction. NN models are generally composed of simple circuits of nodes connected to each other (Merghadi et al., 2020), defined with three layers, i.e. input layer, hidden layer and output layer. The structure applied in this paper is as follows: input layer, first fully connected layer, rectified linear unit activation function, final fully connected layer, softmax function and output. The softmax function which was applied to the final fully connected layer is defined as:

$$f(x_i) = \frac{e^{(x_i)}}{\sum_{j=1}^{K} e^{(x_j)}} \tag{8}$$

where: $K$ is the amount of classes of response variables in the final fully connected layer and $x_i$ each input, resulting in probabilistic [0,1] values, i.e. $f(x_i)$. Merghadi et al., 2020 summarize NN as a "black box" method, unable to interpret relations between input and out variables. However, the method is popular in LSA (Reichenbach et al., 2018), with Habumugisha et al., 2022 even testing different settings to develop a method comparison, such as convolutional, deep and recurrent NN models. Furthermore, the successful applicability of different NN variations is found in the work of Lee, 2006, Nefeslioglu et al., 2007,

Pascale et al., 2013, as well as in the LAND-SUITE software (Rossi et al., 2022).

Introducing the concept of bagging and random feature selection by Ho, 1995 and Breiman, 2001, RF are based on decision trees and provides an improvement over bagged trees (James et al., 2013). Bagging is a procedure introduced to reduce the variance of statistical learning methods, particularly useful for decision trees (James et al., 2013). Defining $p$ as the number of total predictors, and $m$ as the number of predictors taken at each split, in bagging, decision trees are built using the expression:

$$m = p \tag{9}$$

compared to RF, where the relation is defined as:

$$m = \sqrt{p} \tag{10}$$

Namely, the algorithm does not consider most predictors in each decision tree in the RF method. As a result, RF tends to produce precise results and has an increasing popularity in LSA from 2010 onward (Merghadi et al., 2020), as depicted in

several papers such as Catani et al., 2013, Wang et al., 2021, Sandić et al., 2023. Moreover, ensemble methods, including RF, are remarked with excellent performance by Merghadi et al., 2020.

SVM is introduced by Cortes and Vapnik, 1995 and Vapnik, 1995, whereas James et al., 2013 describe SVM as an extension of a support vector classifier that can convert a linear classifier into one that automatically produces non-linear boundaries. Moreover, SVM can deal with linearly separable data and linearly non-separable data, whereas points that constrain the width

of the margin, i.e. are closest to the optimal hyperplane, are called support vectors. To better classify most training observations, the support vector classifier allows for a certain amount of observations on the wrong side of the hyperplane or the margin





(James et al., 2013). The SVM uses kernel functions to transform originally non-separable data from two-dimensional into three-dimensional feature space (Ballabio and Sterlacchini, 2012), where the optimal hyperplane is constructed, and later used to classify new data. Binary classification applied in this study, i.e. 0 being stable and one unstable pixels results in better

prediction efficiency than a one-class SVM application (Yao et al., 2008). Also, equation [6] is applied to convert numerical susceptibility values to probabilistic ranging [0,1]. Merghadi et al., 2020 identified SVM starting a continuous increase in usage in landslide susceptibility studies since 2010 (e.g. Yao et al., 2008, Pradhan, 2013, Kavzoglu et al., 2014) and characterized SVM as having above average performance with moderately easy implementation.

### 3.4 Quantitative and Qualitative analysis

A landslide training set containing unstable and stable modelling pixels was examined to determine model evaluation parameters, i.e. fitting performance by determining Cohen's kappa and AUC for False Alarm Rate and Hit Rate metrics. Cohen's kappa and AUC are often used statistical parameters in landslide susceptibility analyses to evaluate the model, e.g. Pourghasemi et al., 2021 and Tyagi et al., 2023. Concretely, Cohen's Kappa value interpretation was defined in Landis and Koch, 1977 as: (i) almost perfect (0.8 – 1), (ii) substantial 0.61 – 0.8), (iii) moderate (0.41 – 0.60), (iv) fair (0.21 – 0.40), (v)

slight (0.00 – 0.20) or (vi) poor (<0.00), and the equation to determine the metric is:

$$\text{Cohen's Kappa} = \frac{2 \times (TP \times TN - (FN \times FP)}{(TP + FP) \times (FP + TN) + (TP + FN) \times (FN + TN)} \tag{11}$$

Where TP are true positives (i.e. landslide pixels classified as unstable), FN are false negatives (i.e. landslide pixels classified as stable), TN are true negatives (i.e. non-landslide pixels classified as stable), and FP are false positives (i.e. non-landslide pixels classified as unstable) (Gorsevski et al., 2006).

In this study, the susceptibility values were split into 100 classes with 0.01 susceptibility intervals, i.e. defining the classification thresholds used for AUC calculation. TP, FN, TN and FP pixels were determined using 0.5 as a probabilistic susceptibility cut-off value for each interval. Moreover, when the ROC curve was defined, the AUC was calculated to further estimate model fitting performance as in Rossi et al., 2010, Bornaetxea et al., 2018, Wang et al., 2021, whereas the Hit Rate and False Alarm Rate values were defined by Fawcett, 2006 as:

$$\text{Hit Rate} = \frac{TP}{TP + FN} \tag{12}$$

and

$$\text{False Alarm Rate} = \frac{FP}{FP + TN} \tag{13}$$

AUC values closer to 1 indicate a perfect prediction, contrary to 0.5 value corresponding to random prediction. An example for describing performance given by the AUC values is as follows: (i) <0.7 poor; (ii) 0.7-0.8 fair; (iii) 0.8-0.9 good and (iv)

>0.9 excellent (Fressard et al., 2014).

Unstable validation pixels are used to derive a Cumulative percentage study area and Cumulative percentage landslide area, defining prediction performance as introduced by Chung and Fabri, 1999 and Chung and Fabri, 2003. The metric is often used





in bivariate LSA approach (e.g. van Westen et al., 2003, Sinčić et al., 2022b, Bernat Gazibara et al., 2023) and is defined as success rate and prediction rate for fitting and predictive performance, respectively.

The standard deviation (SD) maps for susceptibility values are implemented to measure each scenario's stability and method's stability, resulting in qualitatively detecting uncertainty zones. Namely, 11 SD maps are developed for each of the 11 scenarios by observing the five applied methods in each scenario, i.e. to measure deviations among the methods. Furthermore, five SD maps describing criteria are defined by observing 11 scenarios applied for each method, i.e. to measure deviations among scenarios. Considering the probabilistic result of each LSM from 0.0 to 1.0, SD classes are determined with 0.1 threshold

intervals to observe their presence and spatial distribution in the study area. Qualitatively, all classified SD maps are presented to illustrate spatial distribution, which is also measured quantitatively by determining the percentage area of each SD class in every derived map.

By observing all unstable pixels, AUC values were determined for the Cumulative percentage study area and Cumulative percentage landslide area curve. The latter presents AUC for LSM classification, whereas all unstable pixels are the sum of

unstable pixels used for fitting performance (model evaluation) and predictive performance (validation).

Observing both stable and unstable pixels for model evaluation by Cohen's kappa and False Alarm Rate and Hit Rate AUC metrics were chosen due to their equal presence in the model training. On the other hand, measuring only unstable pixels for validation and classification by developing a Cumulative percentage study are and Cumulative percentage landslide area curve was selected to emphasize landslide occurrence exclusively for validation and classification without considering stable pixels.

It should be noted that all AUC values presented in this paper, i.e. fitting, predictive, and classification performance, are expressed in rates, meaning 100 and 50 correlate to 1.0 and 0.5, respectively.

For uniform comparison, LSMs were classified in this paper according to probabilistic susceptibility values by using cut-off values from Bornaetxea et al., 2018, resulting in five susceptibility zones: (i) very low (0.0 – 0.2), (ii) low (0.2 – 0.45), (iii) medium (0.45 – 0.55), (iv) high (0.55 – 0.8), (v) very high (0.8 – 1.0). Quantitatively, for every 54 LSMs, the class area size

is observed, including landslide presence in each class. Furthermore, classified LSMs are qualitatively illustrated with training and validation landslides to observe spatial distribution, including close up views by overlapping them on high-resolution hillshade maps to investigate spatial accuracy, robustness, and pixelization degree.

## 4 Results

### 4.1 Landslide conditioning factors

In this section, differences between the 11 scenarios described for classifying continuous LCFs are explained on elevation LCF map, quantitatively (Table 4, Fig. 4) and qualitatively (Fig. 5). Moreover, a general overview of all LCFs with the number of classes through 11 scenarios is presented in Table 3, as well as class area distribution in Fig. 7. Following the described methodology, all LCFs in S1 and S2 scenarios keep a rather stable number of classes at a 90-101 and 46-51 range, respectively. The shortage of classes in slope and terrain wetness LCFs is due to missing certain values in the original stretched raster. As





intended, with a fixed amount of 10 classes, S8 to S10 scenarios have identical amounts of classes for all LCFs. Regarding the heuristic S3 scenario, the number of classes varies from 7 classes at elevation to 13 classes in proximity to faults. A relatively low number of classes are depicted in S4 to S6 scenarios where continuous rasters have 2 to 6 classes, whereas vector line (buffer) LCFs show a larger span from 2 to 25 classes. Namely, proximity to geological contact has significantly more classes than other LCFs, followed by proximity to streams and faults. An exception is proximity to drainage network LCF, having 2

to 4 classes in S4-S6 scenarios. As intended, the number of classes in each LCF increases from scenario S4 to scenario S6. The S7 scenario has the lowest number of classes in all LCFs, the highest number of classes being four at proximity to geological contact and the lowest two classes in proximity to drainage network, streams and faults LCF. Edge raster values and edge buffer values defined for the S11 scenario representing input data layers files are depicted in Table 3.

**Table 3 Amount of class number (or values) distribution for LCFs in 11 scenarios**

| | | Amount of classes in a LCF (N) | | | | | | |
|---|---|---|---|---|---|---|---|---|
| | | Elevation | Slope | Terrain wetness | Prox. to streams | Prox. to drainage network | Prox. to geological contact | Prox. to fault |
| S1 | | 100 | 91 | 90 | 101 | 100 | 99 | 99 |
| S2 | | 50 | 46 | 46 | 51 | 50 | 50 | 50 |
| S3 | | 7 | 10 | 12 | 12 | 8 | 12 | 13 |
| S4 | | 3 | 2 | 3 | 8 | 2 | 13 | 6 |
| S5 | | 4 | 3 | 3 | 11 | 3 | 20 | 9 |
| S6 | | 6 | 4 | 4 | 14 | 4 | 25 | 12 |
| S7 | | 3 | 3 | 3 | 2 | 2 | 4 | 2 |
| S8 | | 10 | 10 | 10 | 10 | 10 | 10 | 10 |
| S9 | | 10 | 10 | 10 | 10 | 10 | 10 | 10 |
| S10 | | 10 | 10 | 10 | 10 | 10 | 10 | 10 |
| | | Edge raster values | | | Edge buffer values (m) | | | |
| S11 | FROM | 122.15 | 0.71 | 2.14 | 12 | 1.7 | 12.5 | 13 |
| | TO | 439.94 | 80.26 | 19.90 | 1212 | 170 | 1237.5 | 1287 |

To obtain 100 classes for the S1 scenario and 50 classes for the S2 scenario, equal interval values in the elevation LCF are 3.2 m a.s.l., and 6.4 m a.s.l, respectively. Furthermore, the heuristic interval for the scenario S3 is chosen at 50 m a.s.l. Scenarios S4 to S6 are defined with $C_{st}$ curves based on scenario S1 classes, as illustrated in Fig. 4. For elevation LCF, the

curve is defined with two negative segments in which $C_{st}$ values reach roughly –9 and –7 minimum values, respectively. The





positive segment has a steep increase at class 12, leading to a $C_{st}$ value of 16 at class 24, followed by a decrease in value reaching 0 in class 41, followed by below 0 values. Lastly, class 63 and higher have a $C_{st}$ value of 0, indicating no landslide presence. Figure 4 illustrates that when defining scenario S7, the relation in expected and mapped landslides changes twice, defining three classes. Namely, there are more expected landslides in classes 1 to 13 and 41 to 63, whereas from 13 to 41 there

are more mapped than expected landslides. Regardless of having an equal number of classes, the reclassification cut-off values significantly differ for scenarios S8, S9, and S10, as depicted in the example on elevation (Table 4). On the other hand, despite having a different methodology, scenarios S4 and S7 show minimal differences in the cut-off values for the elevation LCF.

**Table 4 Elevation reclassification cut-off values distribution in 10 scenarios**

| Class (N) | Elevation (119.0 – 436.9 m a.s.l.) reclassification cut-off values | | | | | | | | | |
|---|---|---|---|---|---|---|---|---|---|---|
|  | S1 | S2 | S3 | S4 | S5 | S6 | S7 | S8 | S9 | S10 |
| **1** | 122.2 | 125.3 | 150.0 | 153.9 | 153.9 | 138.1 | 157.1 | 152.5 | 150.0 | 148.9 |
| **2** | 125.3 | 131.7 | 200.0 | 249.3 | 179.4 | 153.9 | 246.2 | 178.6 | 167.4 | 171.1 |
| **3** | 128.5 | 138.1 | 250.0 | 436.9 | 249.3 | 192.1 | 436.9 | 202.2 | 183.6 | 187.6 |
| **4** | 131.7 | 144.4 | 300.0 | - | 436.9 | 249.3 | - | 224.5 | 198.5 | 199.7 |
| **5** | 134.9 | 150.8 | 350.0 |  | - | 274.8 |  | 245.7 | 213.4 | 216.2 |
| **6** | 138.1 | 157.1 | 400.0 |  |  | 436.9 |  | 266.8 | 229.5 | 238.4 |
| **7** | 141.2 | 163.5 | >400 |  |  | - |  | 290.4 | 145.7 | 268.3 |
| **8** | 144.4 | 169.8 | - |  |  |  |  | 316.5 | 261.8 | 308.8 |
| **9** | 147.6 | 176.2 |  |  |  |  |  | 351.2 | 287.9 | 363.3 |
| **10** | 150.8 | 182.6 |  |  |  |  |  | 436.9 | 436.9 | 436.9 |
| **11** | 153.9 | 188.9 |  |  |  |  |  | - | - | - |
| **…** | … | … |  |  |  |  |  |  |  |  |
| **49** | 274.8 | 430.6 |  |  |  |  |  |  |  |  |
| **50** | 278.0 | 436.9 |  |  |  |  |  |  |  |  |
| **51** | 281.1 | - |  |  |  |  |  |  |  |  |
| **…** | … |  |  |  |  |  |  |  |  |  |
| **99** | 433.8 |  |  |  |  |  |  |  |  |  |
| **100** | 436.9 |  |  |  |  |  |  |  |  |  |




**Figure 4 Elevation landslide conditioning factor class cut-off values determined for scenarios S1 (a), S2 (b), S4 (c), S5 (d) and S6 (e) based on studentized curve, and for scenario S7 (f) based on landslide density curve**




Considering class area distribution illustrated in Fig. 5 for the elevation LCF map, scenarios S1 and S11 (Fig. 5a, k, respectively) visually show little difference as intended, S1 being a simulation, i.e. approximation of S11, depicting a continuous change from low to high altitudes. With 50 instead of 100 classes, the S2 scenario depicts a somewhat rougher transition through altitude classes (Fig. 5b). Starting at three classes in scenario S4 (Fig. 5d), the second class defined with a positive $C_{st}$ curve trend splits into two, making up four classes in the S5 scenario (Fig. 5e).


**Figure 5 Maps of spatial distribution of elevation landslide conditioning factor classes for 11 scenarios**





Finally, two classes defined with negative $C_{st}$ curve trends split individually into two additional making up a final count of 6 classes for the S6 scenario (Fig. 5f). The S7 scenario is visually identical to S4 due to almost identical cut-off values depicted in Table 3. Scenarios S8, S9, and S10 (Fig. 5h, i ,j) visually represent moderate differences, mainly visible in the areas of highest altitudes, i.e. the northernmost and central parts of the study area defined with classes 9 and 10. The latter classes are predominantly expressed and prevail in scenario S9 compared to scenarios S8 and S10.

Due to significant class number differences, scenarios S1 and S2 have drastically less class area size than other scenarios (Fig. 6) in all applied LCFs. Concretely, in these two scenarios, the maximum class area is found to be around 10% in proximity to geological contact, proximity to drainage network and terrain wetness LCFs. On the contrary, in the elevation LCF, the class area size does not exceed more than 5% in any class. After roughly the 30th class in scenario S2, the class area size does not exceed more than 1%. Similarly, in scenario S1, the classes in the interval from 60 to 100 usually have nearly 0% class area size. In scenarios S1 and S2, elevation, slope, proximity to all streams and terrain wetness LCFs have an increasing trend in class area size, reaching a maximum followed by a decreased trend. Proximity to geological contact, proximity to fault and proximity to drainage network show only a decreasing trend in scenarios S1 and S2, an exception being proximity to drainage network in scenario S1 with a short exchange of increasing and decreasing trends. For scenarios S3 to S10, all LCFs have around 10 to 14 classes, except proximity to geological contact, which has 24 classes. In any case, for all LCFs in scenarios S3 to S7, most area is contained in the first few classes, e.g. up to class 3 or 4. In elevation, slope and proximity to geological contact LCFs, the maximum class area size is roughly 60%, compared to roughly 80% class area size in terrain wetness, proximity to faults, drainage network and all streams LCFs. Generally, in scenarios S3 to S7 the class area size tends to drastically change from class to class, unlike scenarios S8 to S10, which depict slight changes. As methodologically defined, scenario S9 contains the 10% class area trend through all scenarios, whereas scenario S9 has significantly less class area size in classes 7 to 10, compared to classes 1 to 6.




**Figure 6 Graphs of class area distribution for continuous landslide conditioning factors in scenarios S1 to S10**

## 4.2 Landslide susceptibility model evalution

By using five statistical methods and 11 LCF sets derived differently considering the classification criteria of continuous LCFs, 54 LSM zonation are derived in the susceptibility analysis. Model evaluation parameters consider both stable and unstable pixels used for training the model and expose their fitting performance as illustrated in Fig. 7. Namely, 1.0 Cohen's kappa values indicate that all stable and unstable pixels used for training the model were classified as <0.5 and >0.5 susceptibility values, respectively. Furthermore, Cohen's kappa index has a substantial agreement in all 11 scenarios for IV, LR, NN, and

SVM methods, ranging roughly from 0.6 to 0.7. The latter methods follow the same trend in all scenarios, with NN showing


the best performance (0.7), followed by LR (0.65), and finally, SVM and IV (0.6). RF method showed perfect agreement with 1.0 or near 1.0 values in S1, S2, S3, S6, S8, S9, S10, and S11 scenarios, and drastically lower, i.e. 0.7 in the S7 scenario. Scenarios S4 and S5 in RF performed slightly worse but still with almost perfect agreement, having Cohen's Kappa values 0.88 to 0.96, respectively. Regarding False Alarm Rate and Hit Rate AUC values, the RF method follows the behavior of

Cohen's kappa values, having excellent performance and nearly identical trends as Cohen's kappa through 11 scenarios. IV, LR, and SVM methods depict nearly identical AUC values in all 11 scenarios, ranging from roughly 85.5 to 88.5. In scenarios S4, S5, S6, and S7, the latter methods show values closer to 85.5, whereas the IV method has somewhat poorer results in scenarios S8, S9, and S10. Having the most stable AUC values in all 11 scenarios, the NN method performed excellent with AUC values ranging from 90.5 to 92, which is moderately better than IV, LR, and SVM, yet significantly worse than perfect

agreements in the RF method.

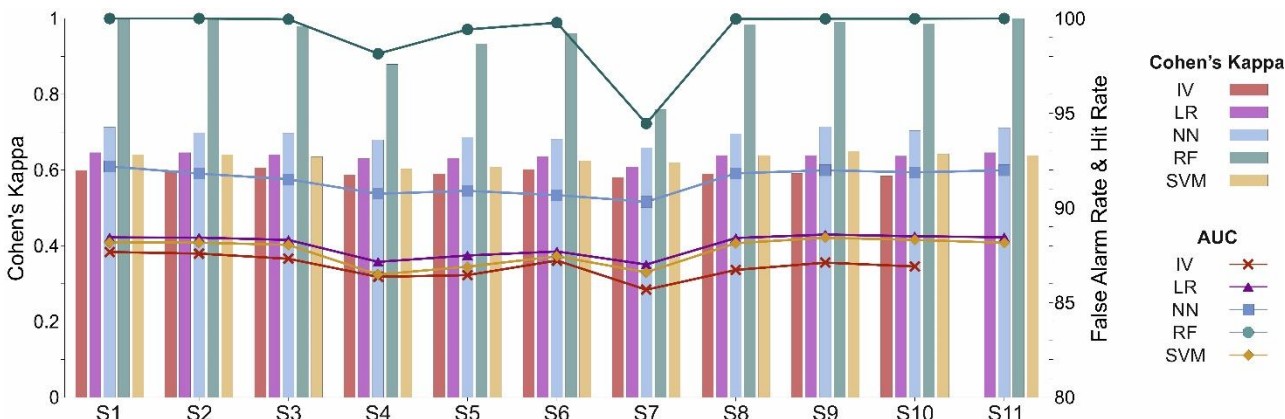

**Figure 7 AUC and Cohen's Kappa quantitative parameters describing fitting performance of 54 derived landslide susceptibility models**

**4.3 Landslide susceptibility model validation**

Unlike results in model evaluation, where studied metrics showed significant differences between five methods, by validating the LSMs, we notice lesser differences with clustered AUC values between methods (Fig. 8). Measuring the predictive performance of LSMs on an independent landslides' dataset results with scenarios S1, S2, S3, S8, S9, S10, and S11 defined with ten or more classes in LCFs as better solutions. In the latter scenarios, the AUC values range from roughly 86.5 to 88.5

for all methods, compared to a decrease in scenarios S4, S5, S6, and S7, where AUC values reach approximately 84 to 87.5. Generally, methods are similar in all scenarios and deviate minimally in scenarios S1, S2, S3, S8, S9, S10, and S11 (approx. 1.5), and moderately in scenarios S4 to S7. Concretively, S11 has proven as a scenario with minimum AUC deviations (<1) between methods, compared to the highest deviations in scenarios S4, S5, S6, and S7 (approx. 3). LR has the best prediction performance in all scenarios with the highest values reaching up to 88.5. Furthermore, LR AUC values are higher up to 3





points compared to RF, which has the least predictive performance, except in scenarios S1 and S2. Having lesser AUC values in most scenarios than LR, NN, and SVM, the IV value method nearly has the best predictive performance in the S7 scenario. On the contrary, LR, NN, and SVM outperform IV in scenarios S8, S9, and S10. Lastly, IV, NN, and SVM have similar predictive performances in scenarios S4, S5, and S6, outperforming RF significantly.

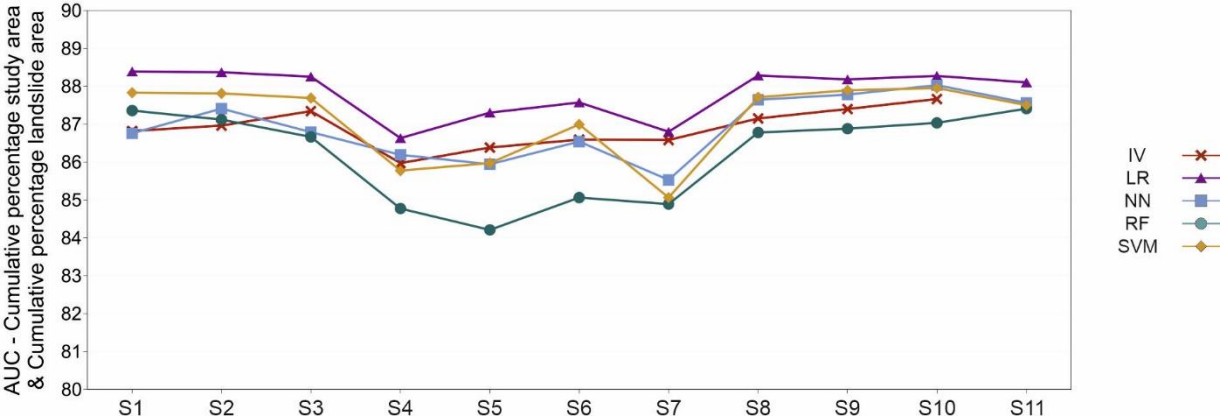

**Figure 8 AUC quantitative parameter describing predictive performance of 54 derived landslide susceptibility models**

**4.4 Landslide susceptibility model classification**

AUC classification values, measuring all unstable pixels as a combination of fitting and predictive performance, show very high values (Fig. 9). IV, LR, NN, and SVM methods show low deviations from scenario to scenario, mainly clustered with AUC values in the range of 84.5 to 89. RF method outperforms the other methods by having AUC values ranging lowest 89 in S7 up to >93 in S11 and S1. After RF, NN proved to be an alternative, having slightly higher AUC values than IV, LR, and SVM, which differentiate minimally in each observed scenario. All methods show lower AUC values in scenarios S4, S5, and S6, with a minimum in scenario S7.

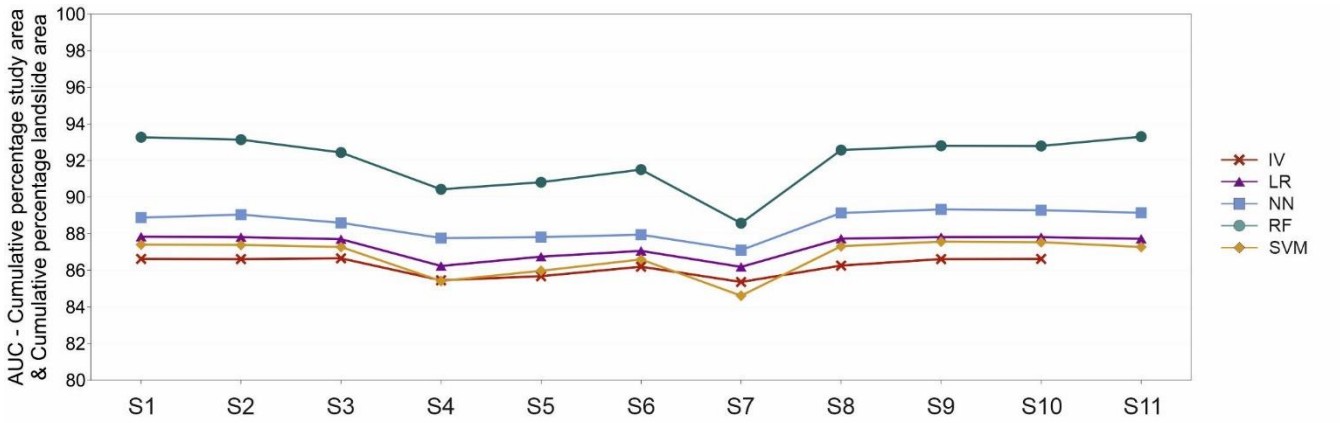

**Figure 9 AUC quantitative parameter describing classification performance of 54 derived landslide susceptibility models**





In scenarios S1, S2, S3, S8, S9, S10, and S11, all methods show AUC values higher than 86. Lastly, the RF method is the only method significantly influenced by the number of LCF classes, with significantly lower AUC values in scenarios S4, S5, S6, and S7.

To identify classification parameters, derived LSMs are firstly classified in zones according to probabilistic susceptibility values as follows: (i) very low (0.0 – 0.2), (ii) low (0.2 – 0.45), (iii) medium (0.45 – 0.55), (iv) high (0.55 – 0.8) and (v) very high (0.8 – 1.0). Generally observing the susceptibility zones, noticeable differences are distinguishable among methods rather than scenarios (Fig. 10). All methods show <10% class area in a very high zone, with RF, SVM, and IV often having <5%. All 54 LSMs have a medium susceptibility zone smaller than 10%, with RF having a minimum with as small as 2%. Moreover,

the RF method has the smallest area in high and very high susceptibility zone. Class area percentage changes are most significant in very low and low classes, whereas very high, high, and medium classes show low differences in all methods and most scenarios. IV and SVM methods tend to have fewer differences between low and very low classes, i.e. having almost equal areas in both zones. RF has the most percentage area in the very low class, followed by NN and LRM. However, RF, NN, and LR have nearly similar class area sizes in the low class. Observing very low and low zones as one, IV, LR, NN, and

SVM show a lot of similarities, unlike RF, which has exceptionally high values, reaching >85% area for the cumulative area of the two classes in most scenarios. Similarly, very high and high zones contain less than 10% of the map area in the RF method, whereas the values reach around 20% for the other methods. In other words, the most significant differences are depicted individually in the relation between very low and low zone and very high and high zone. For the latter, the RF method favors having an extremely large very low zone and an extremely small very high zone. On the contrary, IV and SVM tend to

have rather large low and high zones. NN has around 10% smaller very low zone compared to RF, followed by LR with 10% smaller than NN. Lastly, IV and SVM have around 40% of the map area in the very low zone.

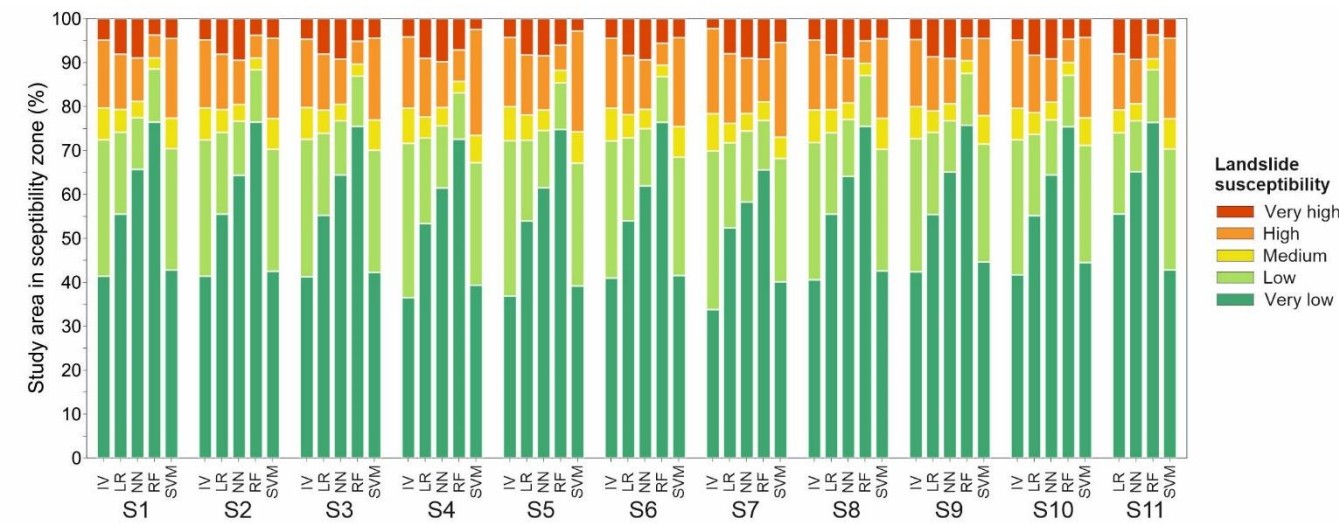

**Figure 10 Landslide susceptibility zone area distribution graph for 54 derived landslide susceptibility models**

The sum of landslide presence in very high and high susceptibility zone are extremely similar in all 54 LSMs (Fig. 11). However, observing the two zones individually, noticeable differences are depicted. Namely, LR, NN, and RF methods have higher landslide presence percentages in very high zone resulting in lower presence in high zone, whereas on the contrary, IV and SVM methods have less landslide presence in very high zone and higher landslide presence in the high zone. Concretely, IV and SVM have around or less than 35% landslide presence in the very high zone, compared to other methods, which often

have >50%. The minimum is in the IV method scenario S7, and SVM method scenarios S4 and S5 at around 20% or less landslide presence in very high susceptibility zone. RF has the maximum landslide presence in the very high zone in each scenario, valuing on average 62%, followed by NN (approx. 56%) and LR (approx. 51%). Observing landslide presence in very low, low, and medium zones, minimal differences are noticeable, an exception being IV and SVM methods, which stand out with extremely low landslide presence in very low susceptibility zone. Moreover, the RF method has the highest amount

of landslide presence in the very high susceptibility zone in each scenario and the highest landslide presence in the very low zone, reaching up to 5%. Interestingly, the average landslide presence area in the medium susceptibility zone is 5.5%, i.e. rather low.

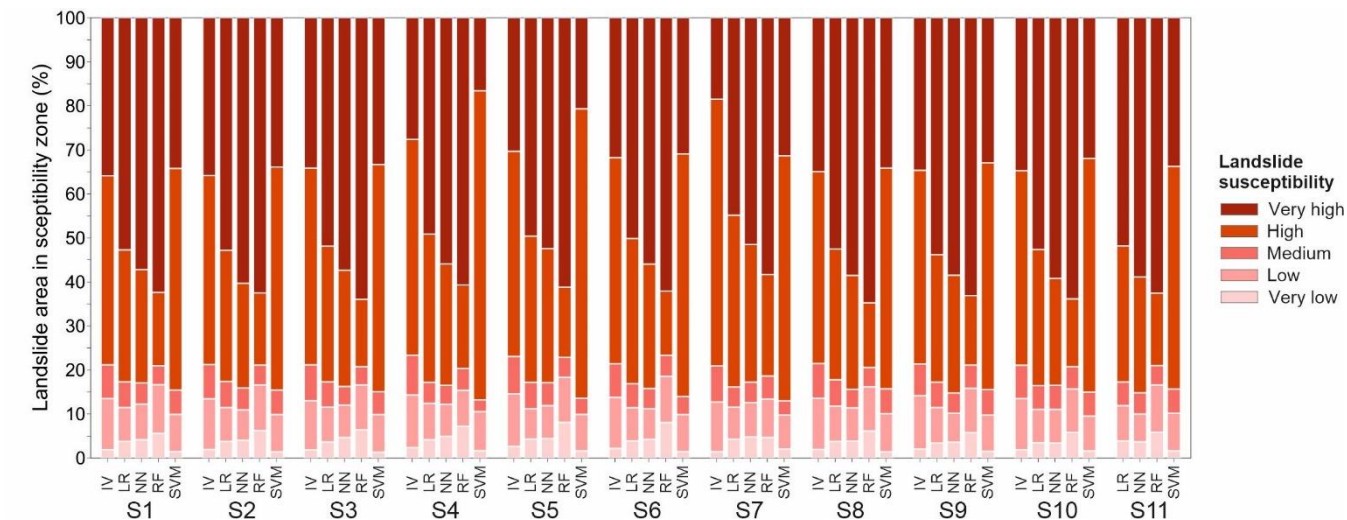

**Figure 11 Landslide area distribution in landslide susceptibility zones graph for 54 derived landslide susceptibility**
**models**

### 4.5 Model variability and qualitative assessment considering LCF classification criteria

11 classified SD maps are illustrated in Fig. S1 for each LCF classification scenario, representing differences in susceptibility values between the five applied methods. Namely, classes of higher SD values, indicate higher uncertainties in the LSMs. The SD maps are visually similar, indicating low differences in susceptibility values from scenario to scenario. 0.0-0.1 SD class is

most represented in all scenarios, with area presence ranging from minimum approx. 50% in scenario S5 to >65% in scenario





S7. 0.1-0.2 SD class and 0.2-0.3 SD class value are equally represented in all scenarios, their area percentage being approx. 35% and 5-10%, respectively. An exception is S7, where the 0.2-0.3 SD class has less than 5%, and in S11, where the 0.1-0.2 SD class has <30% area. Qualitatively observing spatial distribution, SD class distribution does not differ significantly from scenario to scenario. An exception is scenario S11, where the 0.0-0.1 SD zone has clustered areas, unlike dispersed and likely

pixelized, i.e. distributed into smaller zones as visible in other scenarios. Surprisingly, scenarios S7 and S11, which are methodologically opposite considering the number of classes, have a high >60% map area in the 0.0-0.1 SD class. Moreover, S7 stands out by having the least area in the 0.2-0.5 SD range, i.e. having >95% of the area in SD values <0.2. If they are present, 0.3-0.4 and 0.4-0.5 SD classes appear in the same locations in all scenarios. Scenario S11 stands out with the most area in the 0.3-0.4 SD class, but still <1% of the study area. Generally, the 0.3-0.5 SD range is least present and neglectable in

all scenarios when compared to other classes.

Having the best predictive performance, the LR method was chosen for displaying the qualitative spatial distribution of susceptibility zones through 11 scenarios. Namely, the study area view for scenario S1 and 11 close up views for all 11 scenarios are illustrated in Fig. S2a and Fig. S2c-m, respectively. The close up views were selected based on the locations where higher SD values of landslide susceptibility were detected, leading to a better understating of the uncertainty.

By observing the 11 close up views, the differences can be observed by comparing two groups, defined by the number of classes in LCFs used to derive a LSM, and scenario S7 as a standalone. Namely, the first group consists of scenarios with fewer classes in LCFs, i.e. S4, S5, and S6. Scenarios with mainly ten or more classes comprise the second group, i.e. scenarios S1, S2, S3, S8, S9, S10, and S11. The first group is less pixelized, with a low amount of standalone pixels and highly expressed susceptibility zones, unlike the second group, which has heterogeneous and pixelized small zones. Furthermore, the first group

has less high susceptibility class presence in the middle valley stretching north to south. Considering the north-eastern slope, the first group has it classified as a homogeneous high susceptibility zone, compared to a combination of medium and high susceptibility zone in the second group. Both groups classify the eastern and western slopes of the middle valley as a combination of high and very high susceptibility zones, covering most of the landslide area. Scenario S7 has a narrow, very high susceptibility zone stretching in the eastern slope of the middle valley. Moreover, unlike in other scenarios, a wide medium

susceptibility zone in the upper parts of the slope is present. In the northwestern area, scenario S7 has a rather large, very high susceptibility zone. Scenarios S9 and S10 depict both very low and low susceptibility zone in the central valley, unlike other scenarios. Moreover, the latter scenarios have an extremely narrow, very high susceptibility zone in the eastern slope of the central valley, similar to scenario S7. All scenarios except scenario S7 differentiate minimally in the places of close proximity to buildings and roads. Also, differences between susceptibility zone presence on landslide training or validation sets are

neglectable, as both are mainly included in the very high susceptibility zone, or alternatively, to a lesser extent, in the high susceptibility zone.





### 4.6 Model variability and qualitative assessment considering statistical methods

Examining SD maps of susceptibility values for each method individually (Fig. S3), uncertainty through 11 scenarios is measured for each method. Surprisingly, the IV, LR, and SVM methods show minimal SD values for most of the study area.
Namely, the SVM method has 99% of the study area in the 0.0-0.1 SD zone, whereas IV and LR methods have <5% of the study area in the 0.1-0.2 SD zone. In addition, SVM and IV methods display noticeable uncertainty in the south, and southeastern part of the study area, whereas the LR method has them distributed through the study area. Conversely, RF and NN methods have significantly less area in the lowest SD class. For example, RF has >70% area in the 0.0-0.1 SD class followed by >20% in the 0.1-0.2 zone, whereas the NN method has >60% and >30% study area in the mentioned SD zones,
respectively. Finally, both methods have roughly 5% study area in the 0.2-0.3 uncertainty zone.

Observing the spatial distribution of LSMs in the entire study area (Fig. S2a and Fig. S4a-d), the RF method stands out with the most present very low susceptibility zone. Moreover, low, medium, and high susceptibility zones are poorly expressed, whereas very high susceptibility zones are located in landslide areas and nearby surroundings, mainly steep slopes. The NN
method depicts similar spatial distribution, with somewhat larger area of very high, high, and medium susceptibility zones. Unlike RF, which classified most slopes without landslides as very low susceptibility, IV, LR, and SVM methods distinguish the remaining slopes in low, medium, and high susceptibility zones. IV method has a poorly expressed very low susceptibility zone, with a relatively gentle transition between the susceptibility zones.

Considering the close up view comparison (Fig. S4e-h), NN stands out with a very high susceptibility zone mostly present in the western slope of the extent. The SVM method has the slope classified mainly as a high susceptibility zone and partially very high, unlike IV, where the slope is pixelized, and all susceptibility zones are present. On the contrary, the RF method has the slope classified mostly as low and very low susceptible. In a central area with roads and buildings, all methods classify as very low. Instead, SVM has a combination of low and very low susceptibility. Overall, the SVM method on a close up view
shows minimal pixelization, i.e. depicting zones rather than pixel shaped transfer from class to class. RF method proves its poorest predictive performance, with a substantial amount of visible validation landslides in low or very low susceptibility zones. NN and RF methods have the eastern part of the close up view classified as predominantly very low susceptibility, unlike IV and SVM, which have it as very low and low, with small portions of medium susceptibility. RF method certainly has the least very high susceptibility zone presence.





## 5 Discussion

### 5.1 Discussion regarding LCF classification criteria

Considering the quantitative results in Sect. 4.2, 4.3 and 4.4 from a perspective of 11 scenarios for continuous LCFs classification, moderate differences are found in fitting performance and low in predictive and classification performance. However, the noted differences occur repeatedly in scenarios S4 to S7 and are represented by poorer performance in all studied metrics. A likely reason behind the results is too few classes in the LCFs. Concretely, often showing extraordinary results, scenario S7 has four or fewer classes in the continuous LCFs. On the contrary, opposite classification criteria with a large number of classes, such as scenarios S1 to S3 and S8 to S10, result in satisfactory results in all scenarios and are highly similar. That implies that the closer the classification is to the continuous behavior, the LSM fitting and predictive performance is higher.

Considering the above and the results in the previous Sect. 4.5, noticeable differences between the 11 scenarios can be traced to the number of classes present in LCFs, not to the method applied to create the classes. Consequently, scenarios S4, S5, S6, and S7 can be outlined as the less favorable ones. The latter scenarios show poorer fitting, predictive, and classification performance results. Moreover, qualitatively, they have robust susceptibility zones, i.e. loosing susceptibility information on a large scale which is essential for high spatial accuracy and applicability in spatial planning. On the other hand, robustness can ease the classification process to determine susceptibility zones, if the values related to robustness are spatially accurate. Also, compared to other scenarios, the classification process is relatively time-consuming for scenarios S4-S7. Commonly used and straightforward heuristic, NB, Q, and GI classification criteria applied in scenarios S3, S8, S9, and S10, represent satisfactory options, considering the number of classes defined. With most researchers using around ten classes in different methods, considering results from scenarios S1 and S2, a significantly larger amount of classes can also be applied to reach reliable results.

Lastly, scenario S11 can be suggested as a uniform approach to perform landslide susceptibility modelling. Scenario S11 shares excellent performance with several other scenarios, however, it stands out as the most consistent considering predictive performance with low differences between the five used methods. Similarly, it is the least pixelized in the SD map and has >60% of the map area in the 0.0 – 0.1 SD zone, proving low uncertainties between methods. Moreover, directly applying stretched rasters removes a step in landslide susceptibility modelling often done by researchers, enabling more technical simplicity and reducing the time needed due to avoiding the classification of the input data layers. Considering the bivariate approach, where applying stretched rasters such as in scenario S11 without classes is not possible, we suggest scenario S1 as the closest alternative, i.e., the optimal solution.

It should be stated that the set of prepared LCFs in this study is appropriate for achieving excellent performance for deriving LSMs. Moreover, the study aimed to test the 11 criteria on such a set, however, the relevance and/or importance of each LCF in the LSA was not individually determined. Hence, some of the LCFs may be of poor relation to LSA quality, and classifying them differently is irrelevant. In other words, the 11 classification scenarios would likely present more differences on a limited





amount of highly important LCFs. The latter is addressed by applying the classification criteria uniformly for all continuous LCFs and proposing a uniform solution for a relevant set of LCFs, not limited only to the most significant LCFs. Similarly,

applying a non-representative landslide inventory map would likely cause substantial deviations to the results presented in this study. Future work considering this topic could lead to the idea presented by Yan et al., 2019, where LCFs in all scenarios are mutually combined and tested to develop an optimal solution, which would lead to a tremendous amount of LSMs in this paper and was therefore discarded, rather emphasizing in the investigations to propose a uniform approach.

## 5.2 Discussion regarding different statistical methods

Unlike low differences between the 11 scenarios, the five applied methods showed substantial deviations from each other in different quantitative and qualitative performances as seen in Sect 4.2, 4.3, 4.4. and 4.6. Namely, RF outperforms other methods significantly in model evaluation, followed by the NN method, whereas IV, LR, and SVM differ minimally and show the lowest Cohen's Kappa and AUC values. Furthermore, predictive performance favors LR, whereas RF performed poorly, leaving IV, NN, and SVM in between. Lastly, classification metrics favor RF due to perfect results in model evaluation.

Moreover, unlike IV and SVM, RF has the most landslide presence in a very high susceptibility zone. On the other hand, RF also excels in the area presence of a very low susceptibility zone, having it by far the largest compared to other methods while keeping a low percentage landslide presence.

The quantitative and qualitative analysis applied to compare 54 derived LSMs showed the advantages and disadvantages of the five used methods, justifying the broader approach in this study. In model evaluation, RF often classified the training pixels

perfectly, whereas the predictive performance was significantly lower, leaving an open question to investigate further. Nonetheless, a rigorous approach with an unbiased training dataset and 50 to 50 percent landslide inventory splitting still enabled high predictive performance. Moreover, RF clusters the values close to 0 and 1 susceptibility, leaving low and high susceptibility zones relatively small. However, considering landslide presence in the RF susceptibility zones, extreme landslide density in the very high zone can be observed. Interestingly, the IV method can be regarded as equally usable despite the

modelling using only unstable pixels. Moreover, IV was poorly influenced by LCFs having few classes, similar to LR and SVM. All three methods showed extremely low SD values, indicating that all 11 scenarios are equally suitable for their application considering the resulting susceptibility values. On the contrary, NN and RF are affected by the lack of more classes in LCFs, i.e. should not be used recklessly. Considering the classification parameters, low landslide presence was detected in very high susceptibility zones for both IV and SVM. Despite using 100 classes for LCFs in scenario S1, SVM was the only

method that had well-characterized susceptibility zones in close up views, unlike other methods, which were rather pixelized. NN, and even to a greater extent RF, seem to outperform IV, LR, and SVM considering quantitative metrics. However, the qualitative approach depicted certain drawbacks which should be noted. On the other hand, certain advantages were noted in the IV and SVM methods, despite their poorer performance. Consequently, we suggest using LR as a starting point, being the most stable with the least extraordinary results, whereas for IV, NN, RF, and SVM methods, the classification criteria can

drastically influence the LSM quality, i.e. optimization should be applied as well as considering the LSA's purpose.





Considering that in many LSA comparison papers, the mentioned model evaluation and/or validation metrics reach satisfactory results in the tested scenarios, in this study, we emphasized the classification parameters because: (i) satisfactory quantitative metrics results are expected in model evaluation (fitting performance) and LSM validation (predictive performance) due to

complete input data of high spatial accuracy; (ii) classification parameters take to consideration all identified unstable pixels from the complete landslide inventory; (iii) spatial distribution of classified LSMs allows insight regarding application in the spatial planning system, combined with susceptibility zone relation to elements at risk (e.g. buildings, roads); (iv) classification parameters define relations of all identified landslide presence in each defined susceptibility class, essential for application and defining landslide protocols for LSA.

Putting the differences aside, all methods resulted in objectively excellent LSMs, and confirmed the previous LSA done by Bernat Gazibara et al., 2023 in the study area. However, modelling on large scale and aiming towards application in spatial planning, using a qualitative approach has proven to be of great significance by providing new insights into landslide susceptibility modelling. Without the latter, a complete picture of the used methods would not be acquired. LSA is often carried out to optimize certain parameters (e.g. method, mapping unit, or inventory type) by developing dozens of LSMs which are

further compared. When landslide susceptibility modelling reaches a point where most LSMs have a high fitting and predictive performance, we argue that additional metrics (e.g. qualitative approach and classification parameters) are needed, depending on the scope of the LSA.

## 6 Conclusions

The presented work is based on geomorphologically significant and spatially accurate thematic data for LSMs, making it one

of the rare scientific works that implements the recommendations for LSA by Reichenbach et al., 2018. More precisely, Reichenbach et al., 2018 identified that LSA researchers are more eager to experiment with modelling techniques rather than focusing on acquiring relevant thematic and landslide data. Moreover, existing literature deals with preparing thematic input data in a general way and lacks a uniform approach for continuous LCF classification. This work represents a comprehensive analysis of different classification criteria for continuous LCFs and applies the most commonly used statistical methods

evaluated by several quantitative and qualitative metrics to obtain representative conclusions and novelties regarding large scale LSMs.

We defined 11 classification criteria for the seven continuous LCFs applied in the landslide susceptibility modelling, namely elevation, slope, terrain wetness, proximity to geological contact, faults, streams, and drainage network. Also, two criteria were applied to the categorical aspect LCF, whereas the remaining categorical LCFs, i.e. lithology (rock/type) and land use,

remained constant in all scenarios. Scenarios were defined based on the classification criteria applied in the literature review and/or modifications of the criteria and vary from stretched values, partially stretched classes, heuristic approach, classification based on studentized contrast and landslide presence, and commonly used classification, such as Natural Neighbor, Quantiles



and Geometrical intervals. By applying five statistical methods, i.e., IV, LR, NN, RF, and SVM, 54 LSMs were derived, providing a comprehensive insight into the research issue. Also, a strict landslide sampling approach was used, with 50% of landslide polygons used for training and the other 50% for validation. Each derived LSM was measured by quantitative metrics, i.e, False Alarm Rate and Hit Rate AUC and Cohen's kappa for model evaluation, AUC prediction rate values and SD class percentage area for model validation, and AUC, susceptibility zone percentage area and landslide area presence in susceptibility zones for model classification. The complete set of studied metrics proved to be necessary, pointing out the advantages and disadvantages of used methods and classification scenarios. Considering general performance, all the derived LSMs proved to be reliable considering usually studied metrics, i.e. AUC for fitting and predictive performance.

The analysis presented in the paper resulted in a set of large scale LSMs created from representative and spatially accurate input data including detailed LiDAR-based landslide inventory and high-resolution thematic data. Based on 54 reliable LSMs, specific conclusions with practical application could be made. The landslide susceptibility modelling was done on 21 km$^2$ in the Podsljeme area, using 5 m pixel as a mapping unit suitable for large scale LSA. The study area is characterized by small and shallow landslides and can be representative of similar environments with high-quality input data for landslide susceptibility modelling. The main conclusions and novelties derived from the presented comprehensive large scale landslide susceptibility analysis are following:

(i) Due to using relevant input data with sufficient spatial accuracy, landslide susceptibility modelling performed by any statistical method or any LCF classification scenario in this paper resulted in a highly reliable LSM;

(ii) Any of the suggested scenarios to classify continuous LCFs is appropriate if it resulted in roughly ten or unlimitedly more classes in the LCF, suggesting the higher importance of the number of classes in LCFs, rather than the method of how the classes were created. In other words, a low number of classes in LCFs, such as <5, is likely to perform poorly and should be avoided;

(iii) Applying input data layers as stretched rasters (scenario S11) and line vectors as buffers with >10 buffer zones, simplifies the susceptibility modelling process and provides a uniform solution to preparing LCFs;

(iv) Quantitative classification parameters and uncertainty metrics, as well as qualitative comparison (e.g. close up views to verify spatial accuracy) applied in this study, are necessary metrics to evaluate optimal settings for large scale landslide susceptibility modeling as they depict LSM characteristics unidentified by standard quantitative fitting and/or prediction metrics;

(v) Optimal method selection remains an open question and generally should be considered regarding the final applicability of the LSA, whereas in this study LR method presents the most stable and representative option, and the RF method offers optimal performance when appropriately applied, achieving far better performance;

(vi) NN and RF methods are more sensitive to the LCF classification criteria than IV, LR, and SVM.

However, using the same strategy on a different scale or with an incomplete dataset (either irrelevant LCFs or a non-representative landslide inventory) remains an open question which requires additional research. Moreover, comprehensive
comparison using a variety of parameters provides new insight into derived LSMs. For LSM validation, quantitative metrics are not representative due to the lack of a spatial accuracy component, which is especially important in large scale LSA and was proven in this paper. The studied qualitative metrics are maps of SD classes for model validation, whereas the spatial

distribution of susceptibility zones and landslides in full extent and additionally with elements at risk in close up views are used to describe classified LSMs. Presented results and performed validation of LSMs contribute to the preparation of recommendations, evaluation, and use of LSMs and associated terrain zonations proposed by Reichenbach et al., 2018. The recommended qualitative metrics enable the verification of the LSMs for practical application in spatial planning at the local level because it enables comparison with real environmental conditions and elements at risk visible on close up views and

high-resolution hillshade maps.

**Data availability**

Data supporting the research obtainable from the corresponding author upon reasonable request.

**Author contribution**

MS and SBG conceptualised the draft study design, followed by finalizing the research concept with MR. MS carried out the

analyses and wrote the manuscript under SBG supervision. All authors commented and analysed the results and were included in the discussion, whereas SBG, MR, and SMA fully revised the manuscript.

**Competing interests**

The authors declare that they have no conflict of interest.

**Acknowledgements**

This research has been fully supported by the Croatian Science Foundation under the project methodology development for landslide susceptibility assessment for land use planning based on LiDAR technology, LandSlidePlan (HRZZ IP-2019-04-9900, HRZZ DOK-2020-01-2432). and by Institutional project 311980036 ModKLIZ, co-funded by the Faculty of Mining, Geology and Petroleum Engineering.

A funding statement may be additionally adjusted during the review process.

The authors would like to thank Paola Reichenbach for discussions and useful suggestions during the conceptualisation phase of the research.

We would like to thank the reviewers for their suggestions and comments.





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
