# Peer review of "Comparison of conditioning factors classification criteria in large scale statistically based landslide susceptibility models"

_Natural Hazards and Earth System Sciences, 2024_

## Author Comment (AC1)

Dear reviewer,

Thank you for your valuable feedback and please find our responses below.

**Considering general comments:**

Comment: "The paper is well written, and references are appropriated, it is a valuable contribution to susceptibility model assessment, but a few problems remain. The paper is an attempt to objectivise comparison of methods and the impact of classification of conditioning factors. If from page 14 things are clear, it is more difficult to follow the pages 5 to 13."

Response: Thank you for your positive feedback and comments. Considering pages 5 do 8, we believe necessary and simple information is stated considering study area and material. This is also supported by Table 1, Table 2 and Figure 1 to make it more understandable. We rechecked the manuscript for these pages and removed some less important parts to simplify and shorten these pages drastically. For the same purpose, Table 2 is merged with Table 1.

Considering pages to 9 to 13, we made significant changes as follows. A hard to follow Figure 2 has been simplified and moved to new Section 3.2 (now Figure 3). Moreover, Section 3.1 has been excluded and merged into Section 3.3 in a shortened and reduced form. In a new Section 3.1 (old Section 3.2), we added an additional new Table 2 which represents an overview of the 11 scenarios and makes the preparation of input data more understandable. Furthermore, we believe that Figure 3 which we also additionally modified (following same modification style at Figure 4) helps in understanding the more complex scenarios, whereas simpler scenarios are clearly explained in the text. By these changes, we made the entire Section 3 shorter, simpler and homogeneous.

Comment: "A clear description of the scenario's meaning is necessary as well."

Response: A general description of the scenarios can be found multiple times in the manuscript as "scenarios which vary from stretched values, partially stretched classes, heuristic approach, classification based on studentized contrast and landslide presence, and commonly used classification criteria, such as Natural Neighbor, Quantiles and Geometrical intervals". We believe the point is clear to test scenarios which are mainly used in the literature and vary from completely stretched to having minimal classes (in order to cover the two opposites). Nonetheless, we agree with the comment, and the scenario meanings are found in Section 3.1 individually at each scenario explanation, as well as in summarized form in the new Table 2.

Comment: "The first clear sentence about the relationship between several important elements appears in lines 271-273: "*Furthermore, 54 LSMs were derived using the prepared landslide dataset and 11 LCF sets in the five selected methods, i.e., Information Value (IV), Logistic Regression (LR), Neural Network (NN), Random Forests (RF) and Support Vector Machine (SVM).*", which must appear really earlier..."

Response: Thank you for emphasizing this flaw, we agree. To address the issue, we added a similar, clear explanation earlier, i.e. in line 122 (Introduction).

Comment: "...but even though I count only 7 LCF in table 3 (this discrepancy must at least be introduced in the table legend)."

Response: Table 3 depicts changes in the amount of classes for continuous LCFs, whose classes are indeed changing, unlike categorical LCFs. For that reason the table has only 7 (continuous) LCFs, i.e. missing the 3 categorical LCFs whose classes are constant. This flaw is now explained by adjusting the table title, thank you for highlighting the issue. On the other hand, we note that the information for categorical LCFs is visible in Figure 1c (aspect), Figure 1d (lithology soil/rock type) and Figure 1g (land use).

Comment: "I am sure that all the information, but for the reader (at least for me) it is difficult to go in the paper. I would recommend introducing a figure of a graphical flow chart to explain the relationships between LCF, LSM and Section 3.2, figure 2 being rather difficult to follow (maybe adding image may help). This will greatly improve the understanding of the paper."

Response: Thank you for highlighting the difficulties and providing suggestions. A new Table 2 (as also suggested by reviewer 2) is introduced in Section 3.1 which explains the relations of LCFs and LSMs (scenarios). The Table also enables easier following of the text in Section 3.1. Furthermore, we drastically simplified Figure 2 (now Figure 3) as it included basic and trivial workflow information about susceptibility. This way, the entire Section 3.1 is removed as such and merged with Section 3.3 in a more effective manner to shorten the paper. Moreover, a Figure difficult to follow is avoided while keeping the information present, opening more room for the new suggested Table without extending the paper length.

Comment: "I also recommend carefully rereading Sections 4.2 to 4.6 regarding discussions, there are some redundancies that can be avoided by having some arguments in the discussion, since the paper is already rather long."

Response: Thank you for the suggestion. Significant changes were made to Sections 4.5 and 4.6 as they were most lengthy and shared most redundancies considering the Discussion. Namely, they are excluded by transferring key lines into Discussion and

removing the remaining lines. Minor changes were done to section 4.2 and none to Section 4.3 and 4.4 because we found no redundancies and 4.3 is extremely short. Generally, the modifications in the Results Section shortened the paper significantly without losing important content.

**Considering specific comments:**

Comment: "Don does not forget parentheses for dates of publications."

Response: Thank you for the comment, we rechecked the manuscript and added parentheses where needed.

Comment: "Corominas et al. is 2014 not 2013 (online version)"

Response: Thank you for the comment, mistake corrected.

Comment: "Figures 8 and 9, if I understand well the first is linked area and the second is related on pixels."

Response: Thank you for your questions, we explain as follows: both Figure 8 and Figure 9 are related on pixels, whereas the first (Figure 8) is calculated by plotting the cumulative percentage of landslide validation pixels on the Y axis. The validation pixels are based on previously selected 50% of polygon not used for training, i.e. to measure predictive performance. On the other hand, the second (Figure 9) is calculated by plotting both training and validation pixels on the Y axis, i.e. based on 100% of polygons. Figure 9 serves to measure performance of all unstable pixels, unlike only training pixels (Figure 7) or only validation pixels (Figure 8). Explanation can be found in the Section 3.3 "Quantitative and Qualitative analysis".

---

## Author Comment (AC2)

Dear reviewer,

Thank you for your valuable feedback and please find our responses below.

**Considering general comments:**

"The work proposes an analysis of the impact of classifying the conditioning factors on landslide susceptibility mapping. The authors identified eleven scenarios for classifying the conditioning factors and used five different statistical models to generate a total of 54 susceptibility maps. The results are satisfying (confirming the key role that the investigation of the most appropriate conditioning factors plays in producing reliable susceptibility maps), and the article is well-written. It represents a valuable contribution to the improvement of the technique."

Response: Thank you for your positive feedback.

**Considering specific comments:**

"The article is quite long and could be shortened, particularly by summarizing Chapter 2.2 (Input Data), where there are some repetitions, and the results chapter, which contains many statements repeated also in the discussion chapter."

Response: Thank you for your suggestion. Chapter 2.2 (Input Data) was shortened by removing some unnecessary information, as well as removing Table 2 and merging it with Table 1. Both actions led to the shortening of the article. Significant changes were made to Chapters 4.5 and 4.6 as they were most lengthy and shared most redundancies considering the Discussion Chapter. Namely, they are excluded by transferring key lines into Discussion and removing the remaining lines. Minor changes were done to Chapter 4.2 and none to Chapters 4.3 and 4.4 because we found no redundancies and 4.3 is extremely short. Generally, the modifications in the Results Chapter shortened the paper significantly without losing important content. Significant changes were also made in Chapter 3 to shorten the paper.

"In Chapter 3.2, it would be very helpful for the reader to have an image or summary table to describe the 11 scenarios developed for the classification of the conditioning factors."

Response: Thank you for highlighting the difficulties and providing suggestions. A new Table 1 is introduced in Chapter 3.1 which explains the relations of LCFs and LSMs, and also enables easier following of the text in Chapter 3.1. We note that the entire Chapter 3 has been modified to make it shorter, simpler and homogenous (see the track-changes version).

"In Figure 1.d there seems to be a problem with the legend: "geological contacts" and "faults" seem to be inverted, or am I mistaken?"

Response: Thank you for your comment, the two are indeed inverted. We modified the legend accordingly to resolve the mistake.